# Long-term survival of patients receiving home hemodialysis with self-punctured arteriovenous access

Koji Tomori[1], Tsutomu Inoue[1], Masao Sugiyama[2], Naoto Ohashi[2], Hiroshi Murasugi[2], Kazuya Ohama[3], Hiroaki Amano[1], Yusuke Watanabe[1], Hirokazu Okada[1]*

1 Department of Nephrology, Saitama Medical University, Moroyama, Iruma, Saitama, Japan, 2 Department of Clinical Engineers, Saitama Medical University Hospital, Moroyama, Iruma, Saitama, Japan, 3 Department of Clinical Engineering, Gunma Paz University, Takasaki-shi, Gunma, Japan

* hirookda@saitama-med.ac.jp

**Data Availability Statement:** All relevant data are within the manuscript and its Supporting Information files.

## Abstract

### Objective

To determine the long-term survival of patients receiving home hemodialysis (HHD) through self-punctured arteriovenous access.

### Methods

We conducted an observational study of all patients receiving HHD at our facility between 2001 and 2020. The primary outcome was treatment survival, and it was defined as the duration from HHD initiation to the first event of death or technique failure. The secondary outcomes were the cumulative incidence of technique failure and mortality. Cox proportional hazard models were used to identify the predictive factors for treatment survival.

### Results

A total of 77 patients (mean age, 50.7 years; 84.4% male; 23.4% with diabetes) were included. The median dialysis duration was 18 hours per week, and all patients self-punctured their arteriovenous fistula. During a median follow-up of 116 months, 30 treatment failures (11 deaths and 19 technique failures) were observed. The treatment survival was 100% at 1 year, 83.5% at 5 years, 67.2% at 10 years, and 34.6% at 15 years. Age (adjusted hazard ratio [aHR], 1.07) and diabetes (aHR, 2.45) were significantly associated with treatment survival. Cardiovascular disease was the leading cause of death, and vascular access-related issues were the primary causes of technique failure, which occurred predominantly after 100 months from HHD initiation.

### Conclusion

This study showed a favorable long-term prognosis of patients receiving HHD. HHD can be a sustainable form of long-term kidney replacement therapy. However, access-related technique failures occur more frequently in patients receiving it over the long term. Therefore, careful management of vascular access is crucial to enhance technique survival.

**Funding:** The author(s) received no specific funding for this work.

**Competing interests:** The authors have declared that no competing interests exist.

## Introduction

Home hemodialysis (HHD) allows for more intensive treatment for end-stage kidney disease (ESKD). Intensive dialysis has been shown to improve the quality of life and survival of patients [1–3]. However, it also presents challenges, including vascular access complications [1,4–6], loss of residual kidney function [7], and increased burden on patients and caregivers [8]. Intensive hemodialysis increases the number of access cannulations and overall stress on access, which may lead to a higher incidence of access complications [6]. A previous study showed that daily hemodialysis carried a higher risk of access events than conventional hemodialysis, especially in patients with arteriovenous (AV) access [4]. However, the long-term impact of these access complications on HHD continuity remains unclear because the observation period of the study was relatively short.

Therefore, this study aimed to investigate the long-term prognosis of patients undergoing HHD at our facility. All patients receiving HHD through our program utilized an AV fistula (AVF), and we investigated their survival and predictive factors, especially for those who had used an AVF for an extended duration.

## Methods

### Study design and participants

This single-center retrospective cohort study investigated the treatment survival of patients undergoing HHD and the timing and causes of treatment failure. The predictive factors for survival were also investigated. The study cohort consisted of patients aged 18 years or older who initiated HHD at Saitama Medical University Hospital between January 1, 2001, and December 31, 2020. Patients with incomplete data and those who discontinued HHD within the first three months of initiation were excluded from the analysis. The HHD initiation date was the first day the patient underwent dialysis at home. We followed all patients from the initiation of HHD until death, kidney transplantation, relocation, or the end of the study (December 31, 2021).

The requirement for informed consent was waived due to the retrospective nature of the study. The study was conducted in accordance with the Declaration of Helsinki and the protocol was approved by the institutional review board of our facility (HP 2023–019). The study was registered in the University Hospital Medical Information Network Clinical Trials Registry (registration number: UMIN R000057952).

### Data sources

Data were collected from the electronic medical records, and they included standard demographic data, dates of ESKD diagnosis, HHD initiation and disposition, etiology of kidney disease, comorbidities, and the type of kidney replacement therapy (KRT) received before the initiation of HHD (e.g., in-center hemodialysis, peritoneal dialysis (PD), or kidney transplantation). Biochemical data, such as albumin, C-reactive protein (CRP), and hemoglobin (Hb) concentrations were also recorded at the beginning of HHD. Details about the HHD prescriptions were investigated; they included the number of treatments per week, duration of each treatment session, type of vascular access, and cannulation technique used (buttonhole or rope ladder).

The data for our research were retrieved on June 30, 2023, ensuring that the analysis was conducted with the most current information available.

During and after data collection, the authors did not have access to any information that could potentially identify individual participants. All data were anonymized prior to the

analysis, and personal identifiers were removed to maintain the confidentiality of the participants, in accordance with our facility's ethical standards for handling medical records and archived samples.

## Outcomes

The primary outcome (treatment survival) was defined as the time from initiation of HHD to treatment failure. Treatment failure consisted of two outcomes: death or technique failure, which was specifically defined as the discontinuation of HHD for more than 60 days that was not due to death, kidney transplantation, or relocation. The secondary outcomes included the cumulative incidence of technique failure and mortality. In addition, the timing and causes of technique failure and death were examined.

## Statistical analysis

The baseline characteristics at the time of HHD initiation were summarized using descriptive statistics. Continuous variables are reported as mean (standard deviation) if normally distributed and median (interquartile range); otherwise, categorical data are presented as numbers (percentages). Treatment survival at 1, 2, 5, 10, and 15 years were calculated using Kaplan–Meier estimates. The Gray model was used for the cumulative incidence of technique failure and death because death is a competing risk factor for technique failure. Cox proportional hazards models were used to identify the patient characteristics or comorbidities that could predict treatment failure. All P-values were two-sided, and values of 0.05 or less were considered statistically significant. Statistical analyses were performed using EZR (Easy R) version 1.61, which is a graphical user interface for R (R Foundation for Statistical Computing, Vienna, Austria) [9].

## Results

Between January 1, 2001, and December 31, 2020, 79 patients started HHD. Of these, 77 were included in the analysis, after excluding one patient with incomplete data and one patient who withdrew within 3 months (Fig 1).

## Patients characteristics

The patient characteristics at the time of HHD initiation are presented in Table 1. Of the 77 patients, 65 (84.4%) were male. Their mean age was 50.7 years. Chronic glomerulonephritis was the most common primary disease in 30 patients (39.0%). As comorbidities, 18 patients (23.4%) had diabetes, and 4 (5.2%) had coronary artery disease. The median duration of KRT before HHD initiation was 22 months (minimum: 1 month; maximum: 330 months). Prior to HHD, 39.0% of the patients were receiving in-center hemodialysis; 26% were receiving PD, including combined therapy; and 33.8% initiated HHD directly. At our facility, patient education was provided by specialized staff three times a week, and the median duration for HHD training was 102 days. The median number of dialysis sessions was five per week, and the median dialysis duration was 3.5 hours. The median weekly dialysis time was 18 hours. The mean blood flow rate was 200 mL/min, and the mean dialysis flow rate was 500 mL/min. All access procedures were performed using a native AVF, and all puncture techniques were performed using a rope ladder. All patients were treated with a DBB-27 dialysis system (Nikkiso, Tokyo, Japan) and an MH-500CX reserved osmosis system (Japan Water System). A high-flux dialyzer was used for all patients.

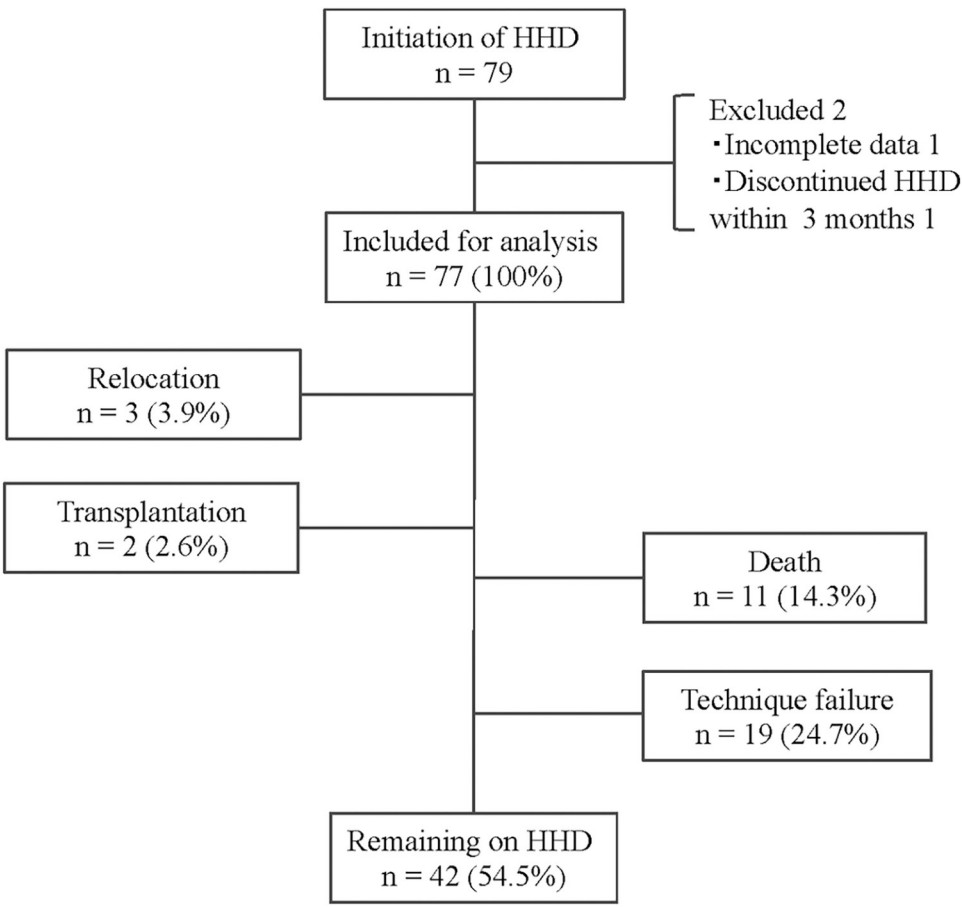

**Fig 1. Disposition of patients enrolled in HHD.** HHD, Home hemodialysis.

### Outcomes

During the median follow-up period of 116 months and a total of 657 patient-years, 3 patients were transferred to another hospital, 2 underwent kidney transplantation, and 42 continued with HHD (Fig 1). Treatment failure was observed in 30 patients, including 19 patients with technique failure and 11 who died. The incidence of treatment failure was 4.57 per 100 patient-years: 2.89 per 100 patient-years for technique failure and 1.67 per 100 patient-years for death (Table 2).

### Primary outcome

Fig 2 shows the treatment survival curves. The treatment survival rate was 100% at 1 year, 83.5% at 5 years, 67.2% at 10 years, and 34.6% at 15 years with a median treatment duration of 167 months.

### Secondary outcomes

Fig 3 shows the cumulative incidence of technique failure and death. In our cohort, the cumulative incidence of technique failure was 0% at 1 year, 9.5% at 5 years, 21.6% at 10 years, and 39.1% at 15 years. The mortality rate was 0% at 1 year, 6.9% at 5 years, 11.2% at 10 years, and 26.3% at 15 years.

**Table 1. Patient characteristics at HHD initiation.**

| Age (years) | 50.7 ± 10.7 |
|---|---|
| Sex n (% male) | 65 (84.4) |
| Cause of ESKD n (%) | |
| CGN | 30 (39.0) |
| DKD | 15 (19.5) |
| HNS | 11 (14.3) |
| PKD | 5 (6.5) |
| NS | 5 (6.5) |
| Other | 5 (6.5) |
| Unknown | 6 (7.8) |
| Comorbidities n (%) | |
| Diabetes | 18 (23.4) |
| CAD | 4 (5.2) |
| Modalities before HHD | |
| In-center hemodialysis | 28 (36.4) |
| Direct HHD initiation | 26 (33.8) |
| Peritoneal dialysis | 20 (26.8) |
| Kidney transplantation | 1 (1.3) |
| KRT vintage, month [IQR] | 22.0 [1.0, 330] |
| HHD Training period, day [IQR] | 102 [73, 161] |
| Dialysis sessions /week [IQR] | 5.0 [4.0, 7.0] |
| Dialysis hours/session [IQR] | 3.5 [2.0, 7.0] |
| Dialysis hours/week [IQR] | 18 [15, 20.3] |
| HDP [IQR] | 100 [40, 196] |
| Type of VA n (%) | |
| AVF | 77 (100) |
| Cannulation technique n (%) | |
| Rope ladder | 77 (100) |
| Albumin (mg/dL) [IQR] | 3.8 [2.1, 4.5] |
| Urea nitrogen (mg/dL) | 62.7 ± 14.5 |
| Creatinine (mg/dL) | 11.2 ± 3.1 |
| Potassium (mEq/L) | 5.6 ± 0.6 |
| Corrected Calcium (mg/dL) [IQR] | 9.1 [6.7, 10.5] |
| Phosphorus (mg/dL) | 5.6 ± 1.2 |
| PTH (mg/dL) [IQR] | 213 [22.2, 1026.9] |
| Hemoglobin | 10.4 ± 1.0 |
| LDL cholesterol (mg/dL) | 96.5 ± 35.3 |
| CRP (mg/dL) [IQR] | 0.11 [0.10, 2.75] |
| $\beta 2$-microglobulin (mg/dL) | 25.1 ± 7.5 |

AVF, Arteriovenous failure; CAD, Coronary artery disease; CGN, Chronic glomerulonephritis; CRP, C-reactive protein; DKD, Diabetic kidney disease; ESKD, end-stage kidney disease; HDP, Hemodialysis product; HHD, home hemodialysis; HNS, Hypertensive nephrosclerosis; IQR, interquartile range; KRT, Kidney replacement therapy; LDL, low-density lipoprotein; NS, Nephrotic syndrome; PKD, Polycystic kidney disease; PTH, Parathyroid hormone; VA, Vascular Access.

**Table 2. Incidence of treatment failure.**

|  | Events | Incidence proportion (%) | Event rates (per 100 patients-year) |
|---|---|---|---|
| Technique failure or Death | 30 | 39 | 4.57 |
| Technique failure | 19 | 24.7 | 2.89 |
| Death | 11 | 14.3 | 1.67 |

## Risk factors for treatment failure

We examined the risk factors for treatment failure using standard Cox regression analysis, considering previously identified variables [3,10–17]. The age at HHD initiation (adjusted hazard ratio [aHR] 1.07, 95% confidence interval [CI] [1.03–1.12], P = 0.002) and diabetes (aHR 2.45, 95% CI [1.1–5.5], P = 0.029) were significantly associated with an increased risk of treatment failure on multivariable analysis (Table 3).

## Cause of death and technique failure

Cardiac disease was the leading cause of death (6 deaths, 50.1%), followed by malignancy (3 deaths, 24.9%) and sepsis (2 deaths, 16.7%). The cardiac-related deaths included two cases of fatal arrhythmias and two cases of heart failure. In addition, four of the five cardiac deaths were sudden deaths.

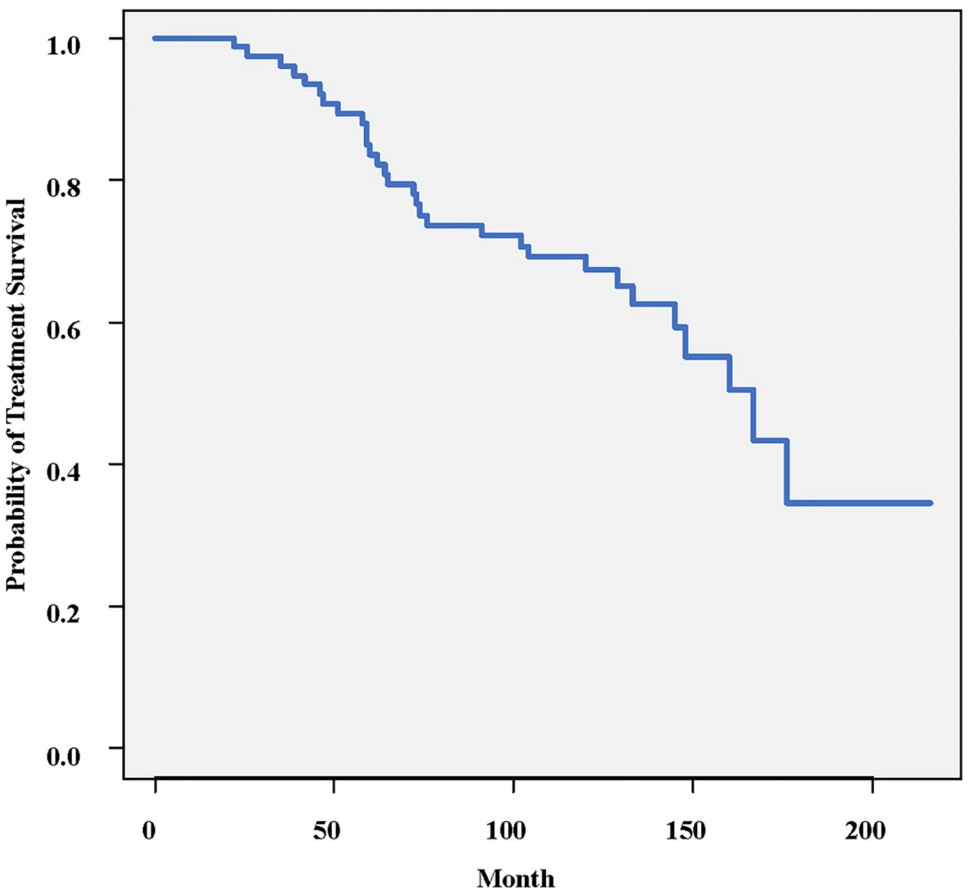

**Fig 2. Kaplan–Meier curve for treatment survival.**

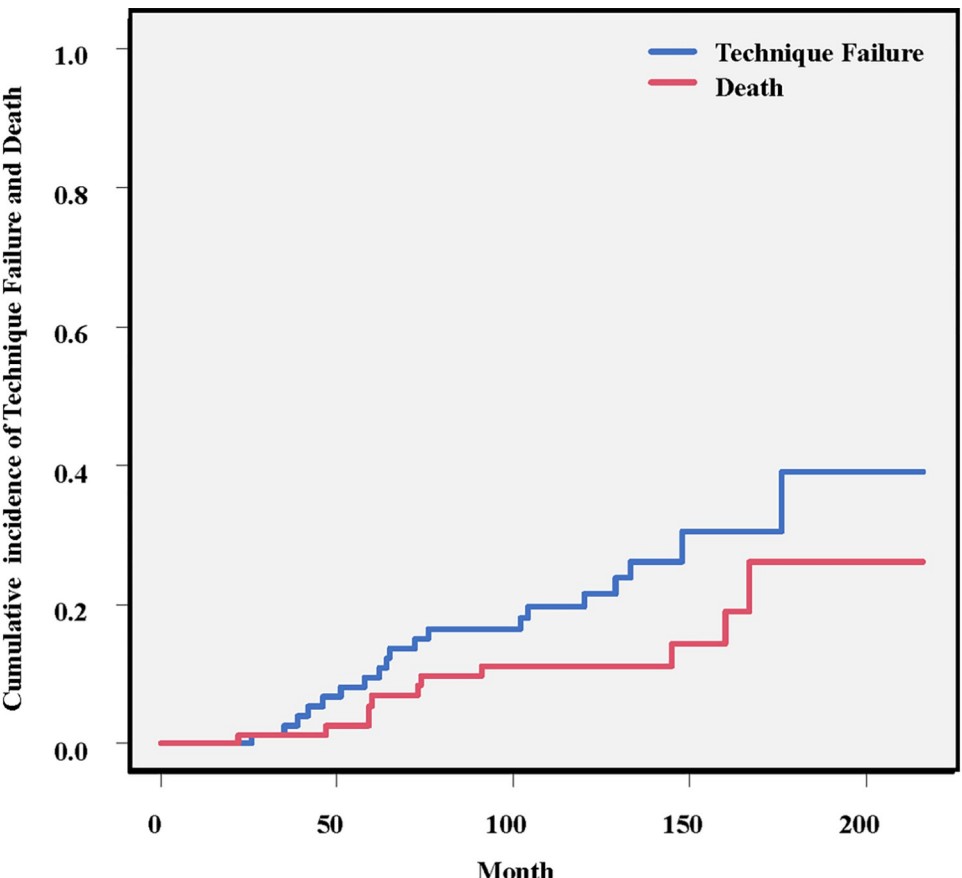

**Fig 3. Cumulative incidence of technique failure and death estimated by the method for competing risk.**

Self-cannulation difficulties were the most common cause of technique failure (6 cases, 31.5%), followed by difficulties with the dialysis technique (3 cases, 16%). When two cases of access failure and one case of access infection were included with self-cannulation difficulties, access-related technique failure accounted for 8 cases, representing 47.3% of all technique failures (Table 4).

## Timing and causes of technique failure

Fig 4 shows the timing and causes of technique failures. Technique failure due to self-cannulation difficulties was more common in patients receiving HHD over long durations, and it

**Table 3. Predictor variables of treatment failure (multivariable model).**

|          | HR   | 95%CI       | p-value a |
|----------|------|-------------|-----------|
| Age      | 1.07 | 1.03–1.12   | 0.002     |
| Female   | 1.75 | 0.57–5.34   | 0.325     |
| Diabetes | 2.45 | 1.10–5.50   | 0.029     |
| CAD      | 2.04 | 0.58–7.10   | 0.325     |

CAD, Coronary Artery Disease; CI, Confidential Interval; HR, Hazard Ratio.

[a] P-values were calculated using the Cox proportional hazards model.

**Table 4. Causes of death and technique failure.**

| Causes of Death | n (%) |
|---|---|
| Cardiac death | 6 (50.1) |
| Myocardial infarction | 2 |
| Lethal arrhythmia | 2 |
| Heart failure | 2 |
| Malignant tumors | 3 (24.9) |
| Malignant lymphoma | 1 |
| Gastric cancer | 1 |
| Lung cancer | 1 |
| Sepsis: | 2 (16.7) |
| Necrotizing fasciitis | 1 |
| Pyogenic knee arthritis | 1 |
| Unknown: | 1 (8.3) |
| Causes of Technique failure | |
| VA-related Technique failure | 8 (47.3) |
| Difficulties with self-cannulation | 6 (31.5) |
| VA failure | 2 (10.5) |
| VA infection | 1 (5.3) |
| Difficulty with dialysis technique | 3 (15.8) |
| Intradialytic hypotension | 1 (5.3) |
| Deterioration of general condition | 2 (10.5) |
| Decline in ADL | 2 (10.5) |
| Poor self-management | 1 (5.3) |
| Decreased motivation | 1 (5.3) |

ADL, Activities of daily living; VA, Vascular access.

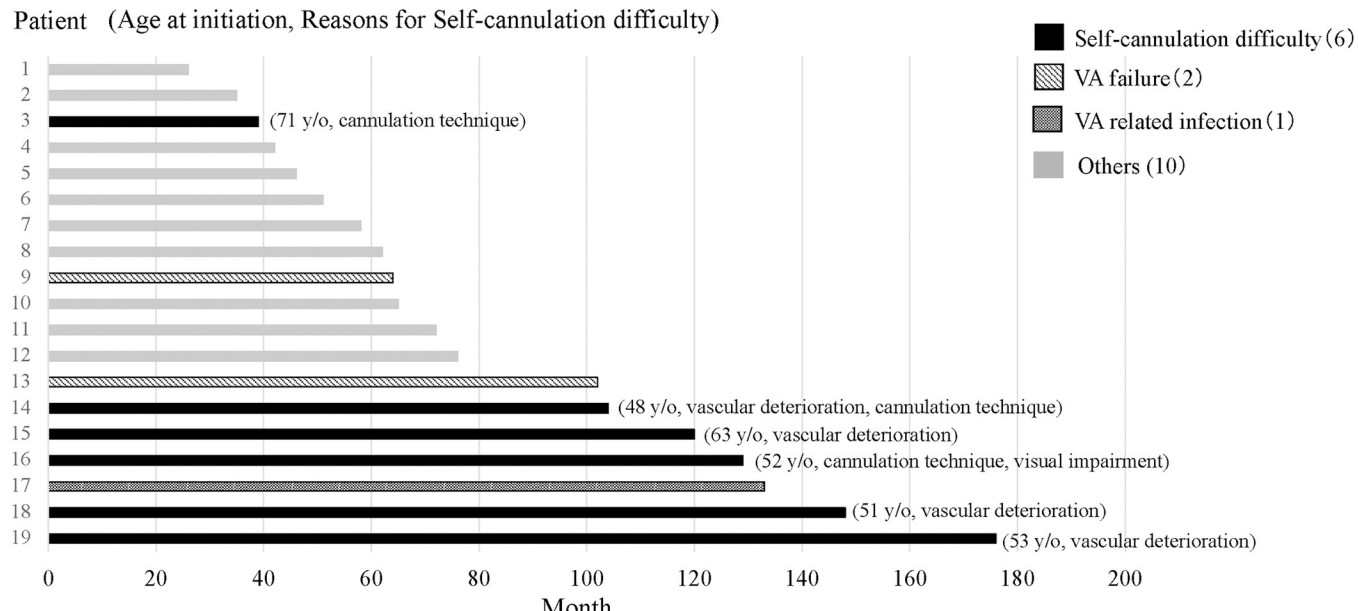

**Fig 4. Timing and reasons for technique failure.** mo, months; y/o, years; VA, vascular access.

occurred after 100 months. Of the five cases with self-cannulation difficulties beyond 100 months, four were primarily due to deterioration at the puncture site. Furthermore, access infections and failures occurred 60 months or more after the initiation of HHD.

## Discussion

In our study on the long-term survival of patients receiving HHD with self-punctured AV access, we identified a favorable prognosis over an extended follow-up duration. This suggests that HHD may be a sustainable form of long-term KRT when effectively managed. Therefore, it is essential to maintain the continuity of HHD, especially in patients with ESKD who require long-term dialysis.

The continuity of HHD is represented by the term "technique survival" or "treatment survival," which has different definitions across studies [3,10–17]. Our study showed favorable results over previous studies (Table 5).

The risk factors for technique failure can be categorized into patient-, center-, and treatment-related [19]. Reported patient-related risk factors include older age [10,14,15,17], cardiac disease [12,16], diabetes [10,12,13,16,17], drug use [13], alcohol [13], and smoking [13,15,17].

**Table 5. Reported HHD continuity rate.**

| | Pauly RP., et al. [10] | Jun M., et al. [11] | Jayanti A., et al. [12] | Seshasai RK., et al. [13] | Pauly RP., et al. [14] | Trinh E., et al [15] | Paterson B., et al [16]. | Ok E., et al. [3] | Our study |
|---|---|---|---|---|---|---|---|---|---|
| HHD modality [18] [a] | Prolonged/ Nocturnal-Frequent | Prolonged/ Nocturnal-Alternate | Short daily: 15% Nocturnal: 30% Alternate: 30% Conventional: 25% | NA | Prolonged/ nocturnal-Frequent | Conventional: 52% Short daily: 14% Slow nocturnal: 3% | Standard-Frequent | Standard-Standard | Standard-Frequent |
| Type of VA AVF (%) | 65 | 92 | NA | NA | 63 | 39 | 44.2 | 87 | 100 |
| Outcome | Composite of Death and Technique failure | Technique failure | Technique failure | Discontinuation from HHD therapy | Technique failure | Technique failure | Technique failure | Composite Death and Technique failure | Composite Death and Technique failure |
| Continuation Rate | | | | | | | | | |
| at 1 year | 95.2 | 90 | 98.4 | 75.1 | 77 | 82 | 92.7 | 93.3 | 100 |
| at 2 years | 88.7 | NA | 95.4 | NA | NA | 76 | 88.6 | NA | 100 |
| at 3 years | NA | 77 | NA | NA | NA | NA | 85.06 | 81.1 | 96 |
| at 5 years | 80.1 | 68 | 88.9 | NA | 57 | 59 | NA | 72.2 | 83.5 |
| at 7 years | NA | NA | NA | NA | NA | NA | NA | 66.7 | 73.6 |
| at 10 years | 52.9 | NA | NA | NA | 27 | NA | NA | NA | 67.2 |
| Risk factors for HHD discontinuation | Age, Diabetes | VA event | Diabetes, Cardiac failure | Diabetes, smoking/ alcohol/drug use, non-listing for kidney transplant, urban residence | Age, additional days of dialysis treatment per week | Age > 65 years Black race, BMI > 30 kg/m2, smoking and small facility size | Diabetes, CAD | longer ESKD duration, higher CVD frequency | Age, Diabetes |

AVF, Arteriovenous failure; BMI, Body mass index; CAD, Coronary artery disease; CHF, Congestive heart failure; CVD, Cardiovascular disease; ESKD, end-stage kidney disease; HHD, Home hemodialysis; NA, not available; SD, Standard deviation; VA, Vascular Access.

[a] HHD modality was classified as follows; for duration, 'Prolonged/nocturnal' >6 h, 'Standard' 3–6 h, 'Short' <3 h; for frequency, 'Frequent' >4x/week, 'Alternate' 3.5 or 4x/week, 'Standard' 3x/week [18].

Consistent with previous studies, our study identified older age and diabetes as risk factors. Therefore, diabetes and advanced age should be carefully considered when selecting and managing patients for HHD initiation.

As a center-specific risk factor, facility size has been suggested to influence technique survival for patients receiving HHD [15,17]. Trinh et al. reported that patients undergoing HHD at larger facilities have a lower risk of technique failure, which is attributed to larger centers having more experience with patients receiving HHD, as well as having more resources and staff available to support both patients and their caregivers [15]. In addition, Pauly et al. reported that differences in care provided by facilities may affect technique survival and patient survival for HHD, and routine nurse home visits are protective against technique failure [14]. Our facility has extensive experience in treating patients with HHD, and specialized nurses conduct home visits every six months. These practices are thought to have contributed to the high rates of treatment continuation. Additionally, considering reports suggesting that the duration of training may be associated with technique survival [20], we included training duration as a covariate in our analysis, but no significant association (aHR 1.003, 95% CI 0.99–1.01, P = 0.15) was observed.

The type of KRT before transitioning to HHD has also been reported to affect technique survival, with evidence suggesting that patients transitioning from PD to HHD have a better prognosis than those who initiate HHD directly [21]. We stratified our analysis by pre-HHD modality, but no significant difference was observed (data not shown). This suggests that previous KRT may not have significantly affected the long-term outcomes of patients who transitioned to HHD.

The type of vascular access and puncture technique have been reported as treatment-related factors associated with technique survival of patients receiving HHD [22–24]. Perl et al. compared patients receiving HHD with a central venous catheter (CVC) with those using an AVF or graft (AVG) and reported that using a CVC is a risk factor affecting the continuation of HHD treatment [22]. Additionally, Verhallen et al. reported that the buttonhole puncture technique may increase the risks of local and systemic infections, potentially leading to the discontinuation of HHD [23]. At our facility, we exclusively used self-puncture AVF for all patients and avoided CVC and AVG. Furthermore, we instructed the patients to use the rope ladder technique for cannulation and avoid buttonhole cannulation. These policies regarding vascular access at our facility may have contributed to the favorable technique survival.

This study identified vascular access issues as the main cause of technique failure, and they were associated with a higher number of patients receiving HHD for more than 60 months. Jun et al. reported that the increased frequency of hemodialysis was associated with higher risks of technique failure and death [11]. Our research showed that a prolonged HHD vintage led to vascular deterioration, making self-cannulation more difficult. In addition, Rousseau-Gagnon et al. suggested that a longer duration of hemodialysis is associated with an increased risk of access-related infections. Patients with longer HHD vintages are more likely to use inappropriate vascular access techniques [24]. Therefore, regular monitoring and patient education are essential to prevent vascular access complications in patients undergoing long-term HHD.

HHD has been shown to improve patient survival [1–3]. In a recent study, Ok et al. compared the survival rates of patients receiving HHD with those receiving in-center HD. They reported an all-cause mortality rate of 3.76 per 100 patient-years for the HHD group and 6.27 per 100 patient-years for the ICHD group [3]. Our study found that the all-cause mortality rate was even lower at 1.67 per 100 patient-years. Our study also found survival rates of 100% at 1 year, 92.5% at 5 years, and 87.4% at 10 years, surpassing those for deceased donor kidney transplantation in Japan, which stand at 95.5% at 1 year, 89.2% at 5 years, and 87.4% at 10

years [25]. In Japan, several patients with ESKD require long-term dialysis due to the low frequency of kidney transplantation. HHD can serve as a bridging therapy between transplantation and viable long-term KRT. Therefore, it is critical to enhance the continuity of HHD.

Intensive HHD may improve survival rates by addressing the key risk factors for mortality in patients undergoing dialysis. These include improved solute clearance, hypertension control, fluid volume management, left ventricular hypertrophy reduction, and maintenance of electrolyte balance. It also reduces the risks associated with long intervals between dialysis sessions [26]. Moreover, the high survival rate of our cohort may be attributed to the low prevalence of heart disease (the leading cause of death among dialysis patients) [27] and the use of AVF in all patients (associated with better survival than central venous catheters and grafts) [28].

This study had limitations. The sample size was small, and the study was conducted at a single center, thus affecting the generalizability of the results. In Japan, non-medical caregivers are not permitted to perform cannulation, meaning that if patients find self-cannulation difficult, they may not continue HHD. Consequently, the applicability of our results may be limited in contexts where medical systems differ. Moreover, Japan lacks HHD-specific dialysis machines, leading to the use of standard in-center machines for HHD. This results in differences in dialysate flow rates and other technique aspects compared to other dialysis practices and healthcare systems globally, further limiting the generalizability of our findings.

Another limitation is the absence of a control group for comparative analysis. Previous studies investigating technique survival also did not set a control group [10–14,16] (Table 5). The primary aim of our observational study was to explore the long-term treatment survival among patients receiving HHD with self-punctured AV access, considering a composite outcome that includes both death and technique failure. We did not include in-center hemodialysis patients as a control group because technique failure, as defined in our study, does not occur in in-center hemodialysis settings.

Despite these limitations, our study was based on reliable and detailed data, and we believe that our findings are invaluable for demonstrating the long-term treatment survival of patients with AVF self-cannulation.

In conclusion, the long-term prognosis of patients receiving HHD through a self-punctured AVF was favorable, and HHD can be considered a sustainable form of long-term KRT. However, access-related technique failures occur more frequently in patients undergoing long-term HHD. Careful management of vascular access is important to improve survival after treatment.

## Supporting information

**S1 File. Data.**
(PDF)

**S2 File. Study protocol.**
(DOCX)

**S3 File. Study protocol (English).**
(DOCX)

## Acknowledgments

We thank the participants for their cooperation. We also express our sincere gratitude to the staff of the blood purification unit at our facility for their invaluable cooperation and support during this study.

## Author Contributions

**Conceptualization:** Tsutomu Inoue, Kazuya Ohama, Hirokazu Okada.

**Data curation:** Masao Sugiyama, Naoto Ohashi, Hiroaki Amano.

**Formal analysis:** Yusuke Watanabe.

**Investigation:** Tsutomu Inoue, Naoto Ohashi, Hiroshi Murasugi.

**Supervision:** Tsutomu Inoue, Hirokazu Okada.

**Writing – original draft:** Koji Tomori.

**Writing – review & editing:** Hirokazu Okada.

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
