## [Decision Letter · Decision Letter 0]

2 Apr 2024

PONE-D-24-07951Long-term survival of patients receiving home hemodialysis with self-punctured arteriovenous accessPLOS ONE

Dear Dr. Okada,

Thank you for submitting your manuscript to PLOS ONE. After careful consideration, we feel that it has merit but does not fully meet PLOS ONE’s publication criteria as it currently stands. Therefore, we invite you to submit a revised version of the manuscript that addresses the points raised during the review process.

As pointed out by both reviewers, the findings of this study are very useful to the nephrology  community and the long-term follow-up of patients is favorable which encourages the use of home hemodialysis, a technique that is still underused globally. I kindly ask the authors to address the minor comments of reviewers, mainly reviewer#1, before the paper is accepted.

 Please submit your revised manuscript by May 17 2024 11:59PM. If you will need more time than this to complete your revisions, please reply to this message or contact the journal office at plosone@plos.org. Please include the following items when submitting your revised manuscript:A rebuttal letter that responds to each point raised by the academic editor and reviewer(s). You should upload this letter as a separate file labeled 'Response to Reviewers'.A marked-up copy of your manuscript that highlights changes made to the original version. You should upload this as a separate file labeled 'Revised Manuscript with Track Changes'.An unmarked version of your revised paper without tracked changes. You should upload this as a separate file labeled 'Manuscript'.If applicable, we recommend that you deposit your laboratory protocols in protocols.io to enhance the reproducibility of your results. Protocols.io assigns your protocol its own identifier (DOI) so that it can be cited independently in the future. For instructions see: https://journals.plos.org/plosone/s/submission-guidelines#loc-laboratory-protocols. Additionally, PLOS ONE offers an option for publishing peer-reviewed Lab Protocol articles, which describe protocols hosted on protocols.io. Read more information on sharing protocols at https://plos.org/protocols?utm_medium=editorial-email&utm_source=authorletters&utm_campaign=protocols.

We look forward to receiving your revised manuscript.

Kind regards,

Mabel Aoun, MD, MPH

Academic Editor

PLOS ONE

Journal Requirements:

2. Please amend the manuscript submission data (via Edit Submission) to include author Naoto Ohashi.

Reviewers' comments:

Reviewer's Responses to Questions

**Comments to the Author**

1. Is the manuscript technically sound, and do the data support the conclusions?

Reviewer #1: Yes

Reviewer #2: Yes

2. Has the statistical analysis been performed appropriately and rigorously? 

Reviewer #1: Yes

Reviewer #2: I Don't Know

3. Have the authors made all data underlying the findings in their manuscript fully available?

Reviewer #1: Yes

Reviewer #2: Yes

4. Is the manuscript presented in an intelligible fashion and written in standard English?

Reviewer #1: Yes

Reviewer #2: Yes

5. Review Comments to the Author

Reviewer #1: Thanks to the authors for their efforts. I have a few points to address regarding this study:

1. The lack of a control group to compare the results with. For example, a control group could consist of in-center hemodialysis patients from the same dialysis facility, matched for age and comorbidities, over a relevant follow-up period.

2. The study included home hemodialysis patients as one group, regardless of their previous kidney replacement therapy. This could potentially affect the results, as patients with prior kidney replacement therapies, especially in-center hemodialysis, or kidney transplantation, may have been exposed to additional factors compared to patients who started home hemodialysis directly.

3. Although patient education before starting home hemodialysis is considered an important issue, there is no mention of the education practices in the dialysis facility. How long was the education provided? What were the means of education?

Reviewer #2: In this retrospective study, the authors describe a very favorable technical survival (defined as death or treatment failure) in their cohort of patients.

Although this was a monocentric study with a relatively small cohort, it was followed over the long term for 19 years, one of the longest reported in the literature. The cohort is well described, attrition bias is low, and the statistical tests used seemed relevant and correctly conducted.

As the authors point out, the transposability of the results may be limited, as the hemodialysis regimen varies with the country and health system. The system used is specific to their center using high dialysate flow hemodialysis rather than low flow. Another special feature is using not only frequent HD, but also relatively long session times.As the authors point out too, the center effect is important in home hemodialysis, and results depend on the center's experience, the type of patients treated and the resources deployed.

However, the very favourable long-term experience with a low drop-out rate is interesting to consider, and the risk factors of age and diabetes on the one hand, and the quality of the vascular approach on the other, are well characterized.

6. PLOS authors have the option to publish the peer review history of their article (what does this mean?). If published, this will include your full peer review and any attached files.

Reviewer #1: No

Reviewer #2: No

---

## [Author Response · Author response to Decision Letter 0]

14 Apr 2024

Mabel Aoun MD, MPH

Academic Editor 

Thank you for pointing this out.

We will resubmit a revised manuscript.

The additional conditions of the revision are as follows:

→We have comfirmed.

2. Please amend the manuscript submission data (via Edit Submission) to include author Naoto Ohashi.

→We have corrected.

3. Please include your full ethics statement in the ‘Methods’ section of your manuscript file.

→We have stated in the Methods section that the requirement for informed consent was waived due to the retrospective nature of our study. Additionally, our protocol has been formally approved by the Institutional Review Board of our facility, under approval number HP 2023-019. 

4. Please include captions for your Supporting Information files at the end of your manuscript, and update any in-text citations to match accordingly.5. Please review your reference list to ensure that it is complete and correct.

→ Captions for the Supporting Information files are included at the end of the text.

5. Please review your reference list to ensure that it is complete and correct. 

→We have reviewed the reference list and confirmed that it is complete and accurate.

We hope that the manuscript is suitable for publication in PLOS ONE.

Sincerely, 

Hirokazu Okada, MD, PhD

---

## [Editor Report · Decision Letter 1]

19 Apr 2024

Long-term survival of patients receiving home hemodialysis with self-punctured arteriovenous access

PONE-D-24-07951R1

Dear Dr. Okada,

We’re pleased to inform you that your manuscript has been judged scientifically suitable for publication and will be formally accepted for publication once it meets all outstanding technical requirements.

Kind regards,

Mabel Aoun, MD, MPH

Academic Editor

PLOS ONE
---

## [Editor Report · Acceptance letter]

10 May 2024

PONE-D-24-07951R1 

PLOS ONE

Dear Dr. Okada, 

I'm pleased to inform you that your manuscript has been deemed suitable for publication in PLOS ONE. Congratulations! Your manuscript is now being handed over to our production team.

Kind regards, 

on behalf of

Dr. Mabel Aoun 

Academic Editor

PLOS ONE